# Chain of Hindsight aligns Language Models with Feedback

**Hao Liu**
UC Berkeley
hao.liu@berkeley.edu

**Carmelo Sferrazza**
UC Berkeley
csferrazza@berkeley.edu

**Pieter Abbeel**
UC Berkeley
pabbeel@cs.berkeley.edu

## Abstract

Learning from human preferences is important for language models to match human needs and to align with human and social values. Prior works have achieved remarkable successes by learning from human feedback to understand and follow instructions. Nonetheless, these methods are either founded on hand-picked model generations that are favored by human annotators, rendering them inefficient in terms of data utilization and challenging to apply in general, or they depend on reinforcement learning, which often suffers from imperfect reward functions and relies on extremely challenging optimizations. In this work, we propose a novel technique, Chain of Hindsight, that is easy to optimize and can learn from any form of feedback, regardless of its polarity. Our idea is inspired by how humans learn from extensive feedback presented in the form of languages. We convert all types of feedback into sequences of sentences, which are then used to fine-tune the model, allowing us to take advantage of the language comprehension capabilities of language models. We condition the model on a sequence of model generations paired with feedback. By doing so, the model is trained to generate outputs based on feedback, while learning to identify and correct negative attributes or errors. Applying our method to large language models, we observed that Chain of Hindsight significantly surpasses previous methods in aligning language models with human preferences. We report significant improvements on summarization and dialogue benchmarks, with our approach markedly preferred in human evaluations. [1]

## 1 Introduction

Large language models have achieved amazing results in natural language understanding (Radford et al., 2018; 2019; Brown et al., 2020). However, in order to ensure that these technologies have a positive impact on society, it is of paramount importance for them to be aligned with human values. One of the most critical elements in achieving this is the use of human feedback. Human feedback allows us to evaluate the performance of such models in a way that is both objective and subjective. It can help to identify issues with accuracy, fairness, and bias, and can provide insights into how the model can be improved, in order to ensure that the model outputs align with societal norms and expectations. Driven by the importance of incorporating human feedback into language models, researchers have been developing and testing various methods for human-in-the-loop systems. These methods aim to make the process of incorporating human feedback more efficient, resulting in models that are able to achieve improved performance and accuracy, while also providing higher fairness and more ethical outputs (Hancock et al., 2019; Perez et al., 2019; Yi et al., 2019; Ouyang et al., 2022; Schulman et al., 2022, *inter alia*).

The successes in language modeling have been largely attributed to the utilization of supervised finetuning (SFT) and Reinforcement Learning with Human Feedback (RLHF) techniques. While these approaches have demonstrated promising results in enhancing the performance of language models on specific tasks, they also suffer from notable limitations. SFT relies on human-annotated data and positive-rated model generation to fine-tune a pretrained language model. However, this approach is heavily reliant on the availability of labeled data, which may entail significant expenses and time investments. Moreover, relying

---

[1] https://github.com/lhao499/chain-of-hindsight

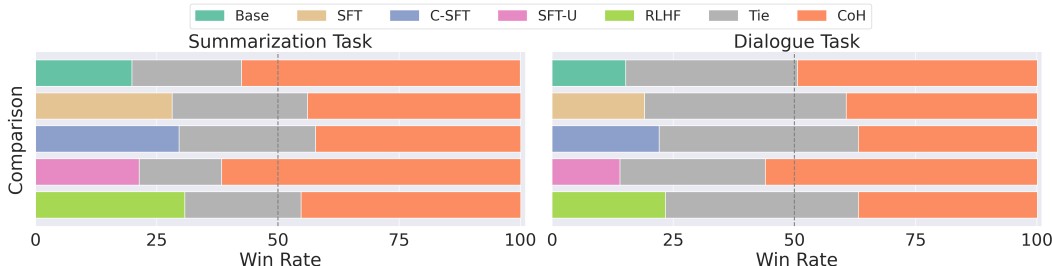

Figure 1: Human evaluation pairwise comparison between CoH and various approaches on the summarization and dialogue tasks. Base denotes the pretrained model, SFT-U denotes SFT with unlikelihood loss, C-SFT denotes conditional SFT. CoH substantially outperform reinforcement learning from human feedback (RLHF) and supervised finetuning baselines.

solely on positive-rated data may constrain the model's ability to identify and correct negative attributes or errors, thus reducing its generalizability to new and unseen data. Alternatively, RLHF enables learning from all data, regardless of feedback rating. Nonetheless, this method requires learning a reward function, which may be subject to misalignment and imperfections (Gao et al., 2023). In addition, the optimization of reinforcement learning algorithms can be challenging, presenting significant difficulties in its application.

In this work, we aim to overcome the limitations of SFT and RLHF by combining their strengths to leverage all feedback, without resorting to reinforcement learning. Our key idea is that humans are capable of learning from rich and detailed feedback in the form of comparisons. Our hypothesis is that by conditioning language models on a sequence of generations paired with feedback and training them accordingly, they can learn to identify and correct errors and negative attributes.

Moreover, prior research has underscored the efficacy of pretrained language models for both in context learning and instruction tuning (Radford et al., 2019; Brown et al., 2020; Wei et al., 2021, *inter alia*). Building upon these insights, we introduce a novel approach: converting all human feedback into a sequence and subsequently finetuning models to comprehend and effectively utilize such feedback. Specifically, we propose finetuning the model to predict outputs while conditioning on one or more model outputs and their corresponding feedback in the form of comparisons to the other outputs.

In essence, our approach finetunes the model by conditioning it to generate outputs while taking into account one or more model-generated outputs and their associated feedback, presented in the form of comparisons to other outputs. During the training phase, the model is given feedback expressions like 'A unhelpful answer' and 'A helpful answer'. It is then tasked with predicting outputs that align more closely with the feedback, such as in the following example: 'How can you explain neural networks to a 6-year-old? A unhelpful answer: {a subpar answer} A helpful answer: {an excellent answer}.' Furthermore, our framework allows for the integration of natural language feedback, such as '{a subpar answer} is a less preferred answer compared with {an excellent answer}', which not only informs the model the preference but also provides additional task-specific guidance. At inference time, when presented with positive feedback indicated by 'A helpful answer', the model is guided to generate the desired outputs, thereby ensuring a preferable behavior.

Our proposed approach enables models to learn from both positive and negative feedback, allowing the identification and correction of negative attributes or errors. We name our method Chain of Hindsight (CoH) as it conditions on a sequence of hindsight feedback. We conducted comprehensive evaluations of our approach in the domains of summarization and dialogue tasks, revealing substantial performance enhancements compared to SFT and its various iterations, as well as RLHF, across both automated assessments and human evaluations.

Our main contributions are twofold: (a) We introduce a novel learning framework, referred to as CoH, which effectively harnesses all available feedback data to enhance model performance without necessitating reliance on RLHF. Notably, our approach CoH maintains the same training objective as pretraining, rendering it straightforward to train and readily scalable; (b)

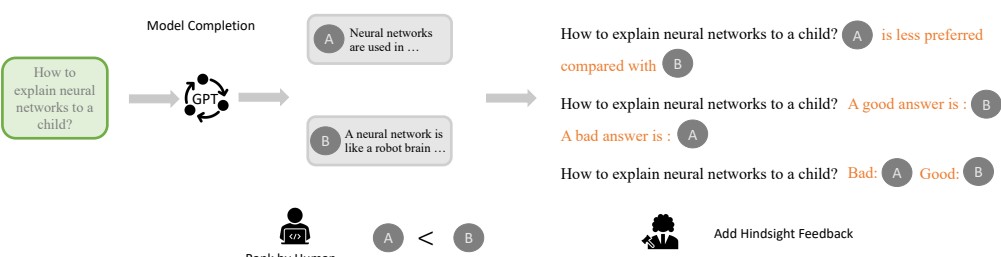

Figure 2: Chain of Hindsight (CoH) turns human preferences into rich and detailed feedback in the form of comparisons. In the diagram, we explain this by showing that a question is being prompted to GPT model. The model then generates a multitude of responses, which are subsequently ranked according to human preferences(*e.g.*, A is less preferred compared with B). Subsequently, we construct CoH sequences by converting human preference into natural language feedback and combine them with the model's outputs. These constructed sequences are then employed in the finetuning phase, aligning with the same objectives as in the pretraining phase.

We conduct extensive experiments to showcase the effectiveness of our method in comparison to existing baselines, including state-of-the-art RLHF methods.

## 2 CHAIN OF HINDSIGHT

Our goal is to improve the performance of a Transformer-based language model by leveraging human-rated data and feedback, and to achieve this, we propose a novel approach that goes beyond conventional SFT methods and RLHF methods.

**Turning all feedback into a sequence.** Our approach aims to take into account all feedback and instructions provided by humans. To achieve this, we present the model with a sequence of model generations, along with corresponding feedback and explanations provided by humans. Our approach uses a conventional Transformer model architecture that is causal and decoder-only, as proposed in the work of (Brown et al., 2020; Vaswani et al., 2017) on attention mechanisms. This means that at each timestep, the model can only attend to the past timesteps and itself. Given a text represented by tokens $\mathbf{x} = [x_1, \cdots, x_n]$, the standard causal language modeling objective is defined to maximize the log likelihood of $\mathbf{x}$ autoregressively: $\log p(\mathbf{x}) = \log \prod_{i=1}^{n} p(x_i|\mathbf{x}_{<i})$. In CoH, we construct $\mathbf{x}$ by combining multiple model outputs with feedback which are then used for instruction finetuning. For instance, when a model is prompted to explain neural networks to a child, it generates multiple responses to the prompt. These responses are then combined together into a sequence and paired with feedback instructions generated based on human ratings. An example is illustrated in Figure 2. During the training phase, the model is presented with both positive and negative feedback denoted as 'Bad' and 'Good', and the model is conditioned to predict outputs that better match the latter feedback such as 'How to explain neural networks to a 6 year old? Bad: {a bad answer} Good: {a good answer}.'. Furthermore, our framework allows for the integration of natural language feedback, such as 'How can you explain neural networks to a 6-year-old? Bad: {a subpar answer} Good: {an excellent answer}', which provides additional task-specific guidance and context. By incorporating a wider range of diverse positive and negative feedback, it further enhances the model's performance. In this study, we opted for templated feedback generated from ratings rather than open-ended feedback from humans in the loop. The feedback type varies depending on the task, we list the contextual natural language feedback in Appendix B.

---

**Natural language feedback examples**

A good summary: {positive}, a worse summary: {negative}
You are a helpful assistant: {positive}, you are an unhelpful assistant: {negative}
A bad answer is {negative}, a good answer is {positive}

---

In theory, one could employ open-ended feedback from humans in the loop. However, for this study, we chose to generate feedback using pre-determined templates based on ratings. During the inference phase, we prompt the model with positive feedback in the form of 'Good' to guide the model in generating favorable outputs.

To enable models to learn from feedback, we require the model to predict each token $x_i \in \mathbf{x}$ that are generated by the model. Loss is not applied on other tokens because it hinders model generation at inference time. This is achieved through masking, which can be expressed as: $\log p(\mathbf{x}) = \log \prod_{i=1}^{n} \mathbb{1}_{O(x)}(x_i) \ p(x_i|[x_j]_{j=0}^{i-1})$, where $\mathbb{1}_{O(x)}(x_i)$ denotes whether token $x_i$ is not part of the hindsight feedback. In other words, it is 1 if $x_i$ is not part of the feedback and 0 if it is part of the feedback. The model is trained to predict each non-feedback token $x_i$ given the previous tokens $[x_j]_{j=0}^{i-1}$.

---

**Algorithm 1** Aligning language models from feedback with Chain of Hindsight.

---

**Required:** Pretrained Language Model M, Human Feedback Dataset D
**Required:** Maximum training iterations $n$
Initialize
**for** $iter = 1$ **to** $n$ **do**
    Randomly sample a minibatch of model outputs and their associated ratings from dataset $D$.
    Construct training sequences by combining sampled model outputs with feedback based on ratings.
    Instruct finetune model $M$ on the training sequences.
**end for**

---

**Training.** We work with a dataset of model outputs and their corresponding human preference, such as positive and negative ratings, from which we sample minibatches of model outputs. To generate hindsight feedback in natural language, we randomly sample a feedback format and incorporate the human ratings. We combine the hindsight feedback and model outputs into a chain of hindsight, which serves as the input for our autoregressive model. The objective is to predict the input sequence autoregressively, and we use cross-entropy loss to optimize the model. We average the loss over each timestep in the last model output sequence. In the regime of human preference learning, the positive and negative data often being similar to each other(*e.g.*, the Anthropic helpful and harmless dataset). Since CoH condition the model on an example when predicting another one, the model can simply 'copy' the example without learning to understand the underlying task. To address this, we randomly mask between 0% and 5% of past tokens during training, which help regularize the model and prevent it from overfitting to the specific examples seen during training (Srivastava et al., 2014; Liu et al., 2022). In order to retain model's performance on general language modeling tasks, we added a regularization term which maximize the log likelihood of the pretraining dataset following prior works (Ouyang et al., 2022). We apply this technique to our method and all baselines in evaluation. Our approach is shown in Figure 2 and the algorithm is summarized in Algorithm 1.

## 2.1 RELATION TO PRIOR PARADIGMS

We discuss the connections of CoH to prior paradigms of learning from preference data.

**Supervised finetuning (SFT).** SFT is a commonly used method for preference learning, involving the use of positively labeled data for finetuning (Ouyang et al., 2022; Schulman et al., 2022). Our approach, however, diverges from SFT by incorporating both positive and non-positive rated data, as well as utilizing feedback input. In comparison to SFT, CoH leverages a broader spectrum of information.

**Conditional SFT.** This method shares similarities with the Decision Transformer model (Chen et al., 2021), which involves conditional training of SFT with feedback serving as prefix tokens. In essence, both CoH and Conditional SFT utilize feedback tokens as conditional input. Nonetheless, the distinction lies in CoH' utilization of a sequence

of feedback-example pairs, enabling our approach to condition on a more comprehensive information when making predictions.

**SFT with unlikelihood.** SFT with unlikelihood introduces an unlikelihood loss on negatively rated data (Welleck et al., 2019; Li et al., 2019) to the traditional SFT framework.

**Reinforcement learning with human feedback (RLHF).** RLHF (Schulman et al., 2022; Ouyang et al., 2022; Stiennon et al., 2020) entails the acquisition of a reward function based on human preferences and the use of reinforcement learning to maximize this reward. In contrast to RLHF, CoH offers a substantially simpler training process, and as our experimental evaluations will demonstrate, it consistently outperforms RLHF in terms of performance.

## 3 EVALUATION SETUP

**Training Datasets.** We use a combination of three datasets for learning from human feedback. The three datasets are:

- **WebGPT.** The WebGPT dataset (Nakano et al., 2021)[2] includes a total of 19,578 comparisons where each example comprises a question, a pair of model answers, and metadata. The answers are rated by humans with a preference score, which helps to identify the better of the two answers.
- **HH.** The Anthropic's Helpful and Harmless (HH) dataset (Ganguli et al., 2022; Bai et al., 2022a) contains human rated dialogues[3]. Each example in this dataset consists of a pair of conversations between a human and a languages model, and one of the two conversations is labeled as preferred by human labelers.
- **Summarization.** The summarization dataset (Stiennon et al., 2020) consists of feedback from humans regarding the summarizations generated by a model[4]. Human evaluators were requested to choose the superior summary from two options presented to them.

**Evaluation Benchmark and Metrics.** We consider both automatic evaluation and human evaluation on summarization and dialogue benchmarks.

- **Summarization Benchmark**. Following prior RLHF works (Stiennon et al., 2020; Nakano et al., 2021; Bai et al., 2022a), we consider automatic evaluation and human evaluation on the TL;DRs dataset (Völske et al., 2017). The original TL;DR dataset contains about 3 million posts from `reddit.com` across a variety of topics (subreddits), as well summaries of the posts written by the original poster (TL;DRs). We use the filtered version provided by Stiennon et al. (2020), which contains 123,169 posts. We evaluate the performance on the validation set. For evaluation metrics, labelers rated summaries for coverage (how much important information from the original post is covered), accuracy (to what degree the statements in the summary are part of the post), coherence (how easy the summary is to read on its own), and overall quality. More details about evaluation dimensions and instructions for human labelers are available in Appendix A.
- **Dialogue Benchmark.** We also evaluate on the validation split of the Anthropic's Helpful and Harmless (HH) dataset (Ganguli et al., 2022; Bai et al., 2022a), where where each example comprises a pair of conversations between a human and a large language model, with one of the two conversations preferred by a human. For evaluating the dialogue, we consider metrics such as helpfulness and harmlessness. A helpful model should follow instructions and infer intention from a few-shot prompt or another interpretable pattern. Since the intention of a given prompt can be unclear or ambiguous, we rely on judgment from our labelers, and the main metric we use is the labelers' preference ratings.

  To collect data for our evaluation, it would be too costly and time-consuming to deploy our finetuned model to chat with humans. Instead, we construct "pseudo" dialogues using positive examples. We replace each model response from a previous dialogue with our model's output, generated by conditioning the model on the human response and past model outputs. We take this approach instead of having humans directly chat with the

---

[2]`https://huggingface.co/datasets/openai/webgpt_comparisons`
[3]`https://huggingface.co/datasets/Anthropic/hh-rlhf`
[4] `https://huggingface.co/datasets/openai/summarize_from_feedback`

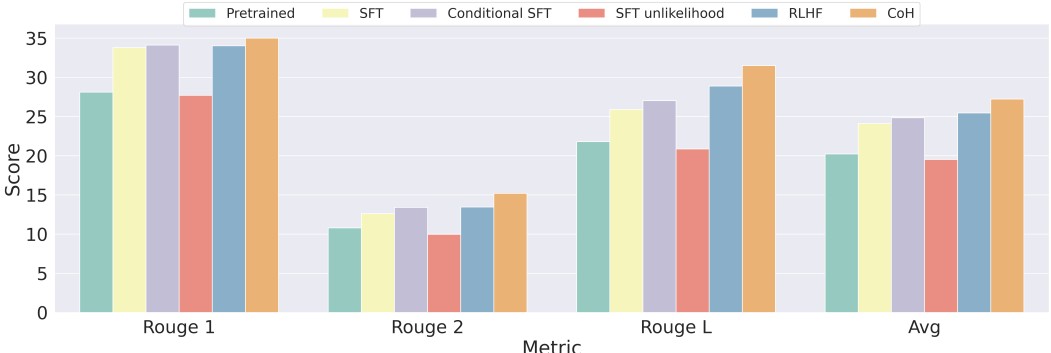

Figure 3: **Evaluation on summarization**. Comparison between RLHF, SFT and CoH. The metrics are ROUGE scores on TL;DR summarization task.

finetuned model to reuse human-generated data, as collecting interactive data can be very costly and is prone to low data quality issues. More details about evaluation dimensions and instructions for human labelers are available in Appendix A.

**Baselines.** Our primary baselines are SFT, SFT with unlikelihood (denoted as SFT-U), conditional SFT (denoted as C-SFT), and RLHF, for connections between them and CoH please refer to Section 2.1. We use GPT-J 6B (Wang and Komatsuzaki, 2021) and OPT (Zhang et al., 2022) as the base pretrained models, while other language models can also be used. Following prior works (Ouyang et al., 2022; Schulman et al., 2022), we adopt the PPO algorithm (Schulman et al., 2017) to implement RLHF baseline. We tune the hyperparameters of PPO and reward learning to obtain the best possible results. To ensure a fair comparison, we carefully tune the training hyperparameters for all other baselines.

## 4 RESULTS

Our main goal in conducting these evaluations is to assess the effectiveness of our proposed methodology, which focuses on summarization and dialogue benchmarks. We conduct both automatic and human evaluations, in order to benchmark our approach against established baselines, including SFT, conditional SFT, SFT with unlikelihood, and RLHF approach (Ouyang et al., 2022; Schulman et al., 2022).

**Evaluation on summarization.** In Figure 3, we present the ROUGE scores of our models on test set of summarization dataset. Our proposed approach, CoH, substantially outperform baselines, including based pretrained model, SFT, conditional SFT, SFT with unlikelihood, and RLHF. Despite the simplicity of our approach, CoH outperforms RLHF across all the metrics. We notice that RLHF performs the second best, with conditional SFT closely follows behind.

To further evaluate the performance of our proposed approach, we conducted human evaluation as shown in Table 1. Base denotes the pretrained model, SFT-U denotes SFT with unlikelihood, C-SFT denotes conditional SFT. We conducted pairwise comparisons between CoH and the baselines because we found that doing so is an easier task for human labelers compared to evaluating multiple

Table 1: Pairwise human evaluation on summarization task.

| | Human evaluation win rate (%) | | | |
|---|---|---|---|---|
| | Base | Tie | CoH | Δ |
| Accuracy | 24.5 | 26.8 | 48.7 | 24.2 |
| Coherence | 15.6 | 18.5 | 65.9 | 50.3 |
| Coverage | 19.6 | 22.4 | 58.0 | 38.4 |
| **Average** | 19.9 | 22.6 | **57.5** | **37.6** |
| | SFT | Tie | CoH | Δ |
| Accuracy | 25.5 | 32.6 | 41.9 | 16.4 |
| Coherence | 30.5 | 25.6 | 43.9 | 13.4 |
| Coverage | 28.5 | 25.4 | 46.1 | 17.6 |
| **Average** | 28.2 | 27.9 | **44.0** | **15.8** |
| | C-SFT | Tie | CoH | Δ |
| Accuracy | 26.7 | 34.9 | 38.4 | 11.7 |
| Coherence | 32.5 | 22.9 | 44.6 | 12.1 |
| Coverage | 29.5 | 26.7 | 43.8 | 14.3 |
| **Average** | 29.6 | 28.2 | **42.3** | **12.7** |
| | SFT-U | Tie | CoH | Δ |
| Accuracy | 18.7 | 17.9 | 63.4 | 44.7 |
| Coherence | 21.8 | 15.8 | 62.4 | 40.6 |
| Coverage | 23.6 | 17.2 | 59.2 | 35.6 |
| **Average** | 21.4 | 17.0 | **61.7** | **40.3** |
| | RLHF | Tie | CoH | Δ |
| Accuracy | 31.8 | 29.5 | 38.7 | 6.9 |
| Coherence | 31.6 | 20.5 | 47.9 | 16.4 |
| Coverage | 28.9 | 21.9 | 49.2 | 20.3 |
| **Average** | 30.8 | 24.0 | **45.3** | **14.5** |

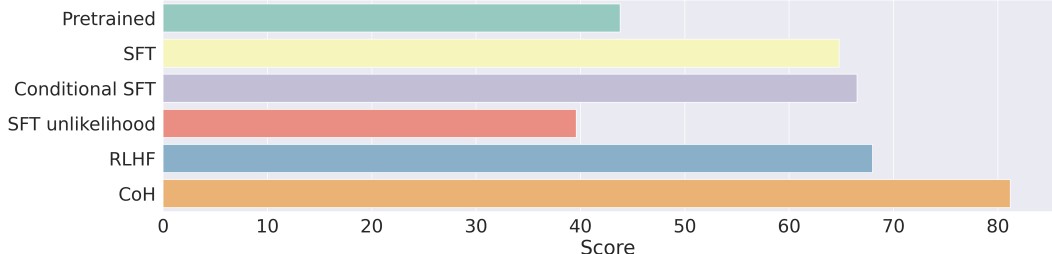

Figure 4: **Evaluation on dialogue**. Comparing CoH with RLHF and SFT baselines. The metric is the accuracy of classifying the preferred dialogue.

options at the same. We hired 75 human labelers who were proficient in English from a third-party platform to provide ratings. In the pairwise comparison, human labelers were presented with two summaries, one generated by the baseline and the other generated by CoH. They were instructed to select the best (or tie) among the two according to the three metrics mentioned above. The metrics are accuracy, coherency and coverage following prior works (Ouyang et al., 2022), we used the same instructions therein, and additional instruct our human labelers to select tie, the full details of human evaluation instructions are provided in Appendix A. Table 1 presents the human evaluation results on summarization task. CoH substantially outperform RLHF and conditional SFT, showcasing the effectiveness of CoH in aligning language models with human preferences.

**Evaluation on dialogue.** We evaluate our method on the HH dataset, by testing its ability to classify which of a dialogue pair is preferred. Figure 4 presents the comparison between baselines and our method. SFT shows substantially improvement over base pretrained model; adding unlikelihood degrades performance which indicates unlikelihood hurts model generation ability; conditional SFT shows improvement over SFT, showcasing the benefit of learning from negative examples; RLHF performs second best and is substantially outperformed by our CoH. The results demonstrate the effectiveness of CoH in learning from preferences. We further evaluate on the dialogue task based on HH dataset. We use the same setting of 75 human labelers and pairwise comparison as in the summarization human evaluation. For this task, we provide human labelers with instructions to evaluate whether the answer is helpful and harmless (Bai et al., 2022a). The results are presented in Table 2. CoH substantially outperform RLHF and conditional SFT, showcasing the effectiveness of CoH in aligning language models with human preferences.

Table 2: Pairwise human evaluation on dialogue task.

| | Human evaluation win rate (%) | | | |
|---|---|---|---|---|
| | Base | Tie | CoH | Δ |
| Helpful | 15.8 | 34.8 | 49.4 | 33.6 |
| Harmless | 14.5 | 35.9 | 49.6 | 35.1 |
| **Average** | 15.2 | 35.3 | **49.5** | **34.4** |
| | SFT | Tie | CoH | Δ |
| Helpful | 19.6 | 45.7 | 34.7 | 15.1 |
| Harmless | 18.6 | 37.4 | 44.0 | 25.4 |
| **Average** | 19.1 | 41.5 | **39.4** | **20.3** |
| | C-SFT | Tie | CoH | Δ |
| Helpful | 21.8 | 46.9 | 31.3 | 9.5 |
| Harmless | 22.4 | 35.2 | 42.4 | 20.0 |
| **Average** | 22.1 | 41.0 | **36.8** | **14.7** |
| | SFT-U | Tie | CoH | Δ |
| Helpful | 13.4 | 31.3 | 55.3 | 41.9 |
| Harmless | 14.5 | 28.7 | 56.8 | 42.3 |
| **Average** | 13.9 | 30.0 | **56.0** | **42.1** |
| | RLHF | Tie | CoH | Δ |
| Helpful | 25.8 | 40.8 | 33.4 | 7.6 |
| Harmless | 20.9 | 38.8 | 40.3 | 19.4 |
| **Average** | 23.4 | 39.8 | **36.9** | **13.5** |

**Language feedback.** We enhance the effectiveness of our approach by evaluating its performance in the context of binary feedback alone, as opposed to the combination of binary feedback and fine-grained language feedback, which is the default setting of our method. We denote this baseline without natural language feedback as CoH w/o LF. To assess the performance of these variations, we conducted a human evaluation task focused on the summarization domain, employing the input of 75 human evaluators. The outcomes, as presented in Table 3, show that both our default approach and our 'w/o LF' variant substantially outperform RLHF. In addition, our findings indicate that the inclusion of natural language feedback enhances the results. Human preference ratings show a 14.1%

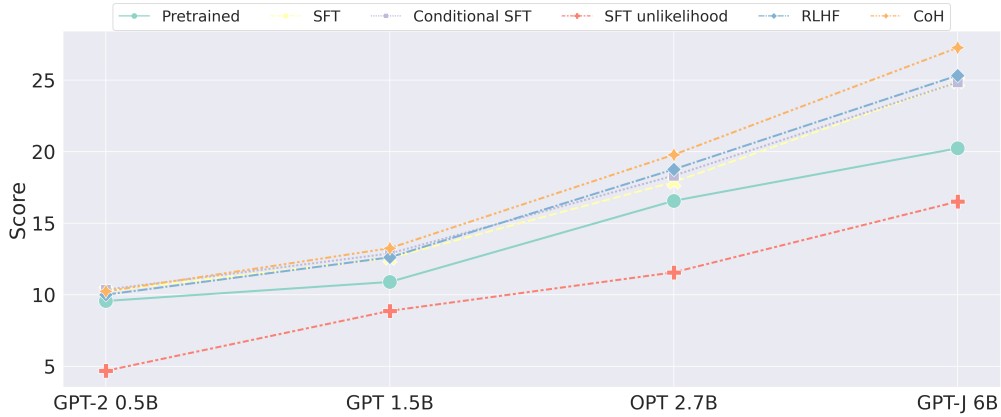

Figure 5: **Model scaling trend**. Comparing CoH with RLHF and SFT baselines on summarization benchmark with different model sizes. CoH outperforms RLHF, showing strong scaling capabilities.

preference for models with language feedback, whereas models without language feedback received an 11.6% preference. The results demonstrate the effectiveness of our CoH framework. Since the framework of CoH offers flexibility to incorporate natural language feedback into training, designing more effective natural language feedback is one of our future directions.

**Evaluation on model scaling trend.** To assess the efficacy of CoH across various model sizes, we conducted a comprehensive evaluation. The findings in Figure 5 demonstrate the impact of varying model sizes on the performance of the CoH method relative to SFT baselines and RLHF. Notably, for smaller model sizes, CoH exhibits a marginal decrement in performance compared to SFT baselines. However, as the model size increases, CoH consistently surpasses all SFT and RLHF baselines and displays a positive scaling trend, indicating its efficacy in enhancing model performance as model complexity increases.

Table 3: Ablation study of natural language feedback on summarization task based on human evaluation.

| Average win rate (%) | | |
|---|---|---|
| RLHF | Tie | CoH |
| 30.8 | 24.0 | 45.3 |
| RLHF | Tie | CoH w/o LF |
| 32.1 | 26.5 | 42.4 |
| CoH w/o LF | Tie | CoH |
| 10.6 | 74.3 | 15.1 |

## 5 RELATED WORK

**Learning from hindsight.** In this paper we explore learning from chains of hindsight with human feedback, an approach that enables a model to learn from errors and revise generations. The key idea of learning from hindsight experience was explored in goal conditioned RL (Kaelbling, 1993; Andrychowicz et al., 2017; Schaul et al., 2015). Andrychowicz et al. (2017) proposes hindsight experience replay (HER) to relabel rewards and transitions retroactively to learn from sparse feedback. While HER relies on reinforcement learning and a distance function to learn from hindsight experience, we propose a new method called CoH that constructs a chain of hindsight experience using human feedback and finetunes the model directly. Our approach offers several advantages over other methods, such as HIR (Zhang et al., 2023), which also makes use of incorrect model outputs. HIR can be seen as a special case of CoH with a length of one chain-of-hindsight. Unlike HIR, which employs a complex training process involving likelihood loss, contrastive loss, and entropy loss, our approach is straightforward and easy to implement. Concurrently, Korbak et al. (2023) studies conditioning on human preference during pretraining and shows improved performance in aligning language models with human preference. Their method is similar to CoH with a length of one chain-of-hindsight. Our work focuses on finetuning pretrained language models while Korbak et al. (2023) focuses on improving pretraining.

**Learning from human feedback.** Prior work have explored using human feedback to improve various tasks, such as summarization (Böhm et al., 2019; Ziegler et al., 2019; Stiennon et al., 2020), dialogue (Yi et al., 2019; Hancock et al., 2019; Bai et al., 2022a;b;

Askell et al., 2021; Scheurer et al., 2022), translation (Kreutzer et al., 2018; Bahdanau et al., 2016), semantic parsing (Lawrence and Riezler, 2018), story generation (Zhou and Xu, 2020), review generation (Cho et al., 2018), evidence extraction (Perez et al., 2019), and instruction following (Ouyang et al., 2022; Bai et al., 2022a). The main techniques behind them can be categorized as supervised finetuning (SFT) or training on filtered human annotations and learning a reward function from human feedback for reinforcement learning, which is often dubbed as RLHF (Christiano et al., 2017; MacGlashan et al., 2017; Lee et al., 2021; Warnell et al., 2017) and has been used to train RL agents without the need for hand-designed rewards. Ouyang et al. (2022) demonstrates improved language model alignment performance by training models with SFT and RLHF using human feedback. Our work belongs to the category of SFT, and differs from SFT in that our method conditions on feedback and can learn from examples without positive ratings. Our method is complementary to RLHF and can be directly combined together for further improvement. Using instructions to provide models with human preference and desired behaviors is demonstrated in Bai et al. (2022b), where models are prompted with a set of statements/principles and are trained with RLHF. In our work, we provide models with a sequence of model outputs and their feedback and train models to generate desired outputs conditioned on feedback/control tokens.

**Instruction finetuning and conditional training.** Finetuning on chain of hindsight using human feedback is akin to instruction finetuning. Driven by the impressive in-context learning ability of large language models, finetuning pretrained models on instructions has been shown to improve language models in many benchmarks (see e.g. Wang et al., 2022; Mishra et al., 2021; Ye et al., 2021; Chung et al., 2022; Wei et al., 2021; Sanh et al., 2021; Zelikman et al., 2022; Huang et al., 2022, inter alia). Mostly the instructions are reformatted examples from NLP benchmarks (e.g. Wei et al., 2021; Chung et al., 2022). CoT prompts (Wei et al., 2022) are widely considered as instructions in prior works (Chung et al., 2022; Wei et al., 2021), specifically in the form of step by step explanations written by humans. In relation to these, our chain of hindsight consists of human written hindsight feedback and ranked model outputs. Conditional training (Keskar et al., 2019; Ficler and Goldberg, 2017; Laskin et al., 2022; Chen et al., 2021; Fan et al., 2018; Lu et al., 2022) explores conditioning the model on some control tokens for controllable generations. In relation to it, CoH generalizes to condition on a sequence of control tokens instead of one control token. By doing so, CoH enables the model to understand the differences between control tokens and their corresponding outputs. Our work suggests a promising direction of using hindsight feedback to construct instructions from model outputs, and can be combined with prior instruction finetuning and conditional training works for further improvements.

## 6 CONCLUSION

In conclusion, we introduce Chain of Hindsight (CoH), which is inspired by how humans learn from rich feedback in the form of comparison. We condition language models on a sequence of hindsight feedback, allowing them to effectively leverage all examples regardless of their preference score. Extensive experiments on summarization and dialogue datasets show that CoH substantially outperform RLHF and other baselines.

**Limitations and Future Work.** Although our method substantially outperform baselines, it does have some limitations that need to be addressed:

- Constructing CoH may result in long sequences, particularly with multiple feedback instances, leading to increased training computational expenses.

- Our work heavily relies on hired human labelers for evaluation due to their higher reliability compared to automated metrics. However, this approach incurs substantial costs, although this issue is not unique to our method.

In terms of future prospects, our CoH-based training from human feedback opens the door to exciting possibilities, such as integrating external environment feedback like unit tests and extending its applicability to various domains. Furthermore, our current focus on learning from hindsight using preexisting preferences paves the way for exploration in online preference learning, enabling iterative model improvements.

## ACKNOWLEDGMENTS

This project is supported in part by Office of Naval Research grant N00014-21-1-2769 and SNSF Postdoc Mobility Fellowship 211086 and ONR MURI N00014-22-1-2773. We express our gratitude to the BAIR communities for their insightful discussions and feedback. We thank Google TPU Research Cloud for granting us access to TPUs.

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

What makes for a good summary? Roughly speaking, a good summary is a shorter piece of text that has the essence of the original – tries to accomplish the same purpose and conveys the same information as the original post. We would like you to consider these different dimensions of summaries:

**Accuracy**
For this axis, answer the question "does the factual information in the summary accurately match the post?" A summary is accurate if it doesn't say things that aren't in the article, it doesn't mix up people, and generally is not misleading.

**Coherence**
For this axis, answer the question "how coherent is the summary on its own?" A summary is coherent if, when read by itself, it's easy to understand and free of English errors. A summary is not coherent if it's difficult to understand what the summary is trying to say. Generally, it's more important that the summary is understandable than it being free of grammar errors.

**Coverage**
For this axis, answer the question "how well does the summary cover the important information in the post?" A summary has good coverage if it mentions the main information from the post that's important to understand the situation described in the post. A summary has poor coverage if someone reading only the summary would be missing several important pieces of information about the situation in the post. A summary with good coverage should also match the purpose of the original post (e.g. to ask for advice).

Table 4: Instructions we provided to human labelers on evaluating summarization quality.

## A  HUMAN EVALUATION INSTRUCTIONS

For our human evaluations, we provide instructions and metrics definition to the human labelers, asking them to select the preferred output. In order to maintain consistency and build upon prior research (Stiennon et al., 2020; Bai et al., 2022a), we adopt their instructions and definitions of helpfulness, usefulness, and other relevant criteria.

Specifically, the instructions employed in our summarization benchmark are derived from Stiennon et al. (2020). Similarly, for the dialogue task, we derive the instructions based on Bai et al. (2022a). Table 4 provides more detail on the specific instructions given to labelers for comparing summaries, and Table 5 lists our instructions for evaluating dialogues.

## B  NATURAL LANGUAGE FEEDBACK

During inference time, we only employ simple positive tokens, while during training, we explored the incorporation of natural language feedback that carries more semantic meaning. This natural feedback is tailored to the specific task and offers increased diversity, as illustrated in Table 6.

## C  HYPERPARAMETERS

All models are trained with the Adam (Kingma and Ba, 2014) optimizer, with $\beta_1 = 0.9$, $\beta_2 = 0.95$, and an epsilon of $1.0e-8$. The batch size for human feedback data is set to 512, while for pretraining data it is set to 2048. The value of $\lambda$ is 1.5, which determines the relative strength of gradients from the human feedback dataset and the pretraining dataset. The pretraining regularization term is computed using the Pile dataset (Gao et al., 2020). Since we applied random past token masking, dropout is not used in our experiments, as suggested by Liu et al. (2022). When finetuning, we combined three human feedback datasets, and the data was sampled proportionally to their size to ensure balance across the datasets.

What makes for a good dialogue? Roughly speaking, a good dialogue is a conversation that is both helpful and harmless. We would like you to consider these different dimensions of dialogues:

**Helpful**

For this axis, assess whether the conversation promotes active listening and open-mindedness. Helpful dialogues typically encourage the exchange of diverse perspectives and ideas. Secondly, look for evidence of empathy and respect in the dialogue; a constructive conversation should foster an atmosphere where all parties feel valued and understood. Thirdly, examine whether the dialogue leads to tangible outcomes or solutions to the issues being discussed. A helpful dialogue often results in progress or resolution, rather than perpetuating conflicts or misunderstandings.

**Harmless**

For this axis, response to questions should not contain potentially harmful information such as "information on building a bomb". Consider the overall tone and language used during the conversation; a helpful dialogue should be free from hostility, personal attacks, or derogatory remarks. A dialogue is harmless if it does not contain any unsafe or potentially harmful information.

Table 5: Instructions we provided to human labelers on evaluating dialogue quality.

Table 6: Examples of Natural language feedback. The task prompts are omitted for simplicity.

| Source | Examples of natural language feedback |
|---|---|
| Summary | a good summary is: {positive} a bad summary is: {negative} |
| Summary | a bad summary is: {negative} a good summary is: {positive} |
| Summary | a good summary is: {positive} a worse summary is: {negative} |
| Summary | a bad summary is: {negative} a better summary is: {positive} |
| Shared | a good response is: {positive} a bad response is: {negative} |
| Shared | a bad response is: {negative} a good response is: {positive} |
| Shared | a good answer is: {positive} a bad answer is: {negative} |
| Shared | a bad answer is: {negative} a good answer is: {positive} |
| Shared | a good answer is: {positive} a worse answer is: {negative} |
| Shared | a bad answer is: {negative} a better answer is: {positive} |
| Shared | good: {positive} worse: {negative} |
| Shared | bad: {negative} better: {positive} |
| Shared | good: {positive} bad: {negative} |
| Shared | bad: {positive} good: {negative} |
| Dialogue | you are a helpful assistant: {positive} you are an unhelpful assistant: {negative} |
| Dialogue | you are an unhelpful assistant: {positive} you are a helpful assistant: {negative} |
| Dialogue | you are a respectful and unbiased assistant: {positive} you are a disrespectful and biased assistant: {negative} |
| Dialogue | you are a disrespectful and biased assistant: {positive} you are a respectful and unbiased assistant: {negative} |
| Summary | give me a good summary: {positive} give me a worse summary: {negative} |
| Summary | give me a bad summary: {negative} give me a better summary: {positive} |
| Summary | let's generate a good summary: {positive} let's generate a worse summary: {negative} |
| Summary | let's generate a bad summary: {negative} let's generate a better summary: {positive} |
| Shared | let's generate a good answer: {positive} let's generate a worse answer: {negative} |
| Shared | let's generate a bad answer: {negative} let's generate a better answer: {positive} |

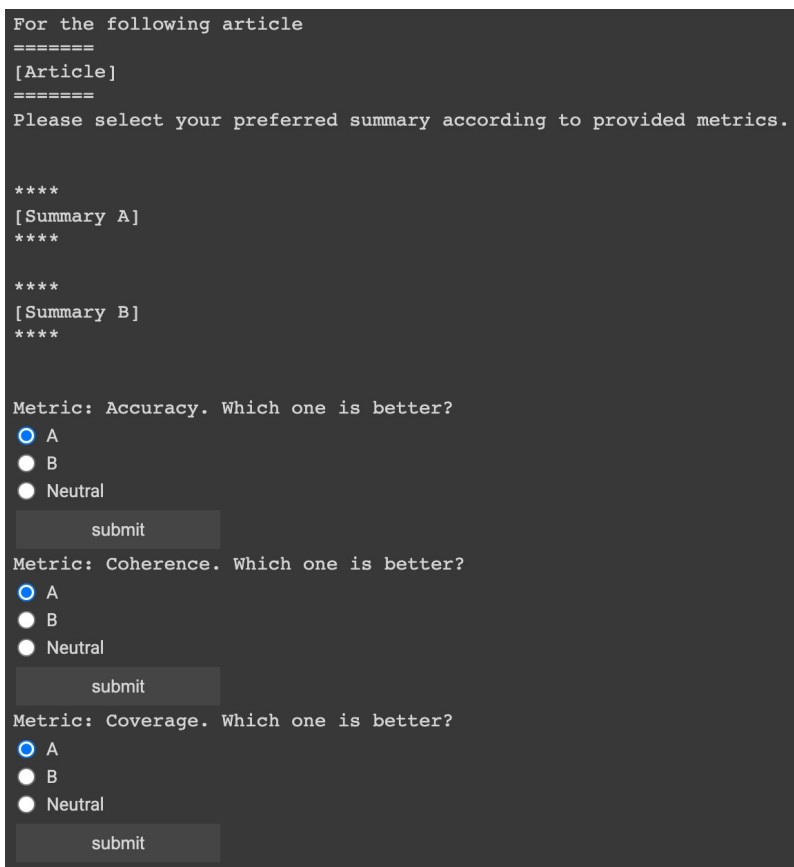

Figure 6: Screenshots of our labeling interface for rating dialog. For each metric, labelers are asked to choose preferred dialog.

Figure 7: Screenshots of our labeling interface for rating summary. For each metric, labelers are asked to choose preferred summary.

## D   HUMAN EVALUATION WEB INTERFACE

In Figure 7 and Figure 6, we show screenshots of our labeling interface, that all of our labelers use to rate data. Labelers can choose the preferred model output or choose tie in cases where two outputs seem to be of similar quality.

# E    ADDITIONAL EXPERIMENTAL RESULTS

## E.1    EVALUATION ON CONTROLLABLE GENERATION

The controllable generation results are presented in Figure 8. The models are provided with three instructions to generate summaries of desired quality. The first instruction asks for a standard summary, while the second and third instructions ask for improved summaries conditioned on the previous summary generated by the model. We compare the performance of CoH with that of the RLHF model. The results indicate that while RLHF performs well in modeling human preferences and generates high-scoring summaries by following the first instruction, it fails to follow the second and third instructions, which implies that it cannot comprehend human intentions. On the other hand, the CoH-trained model is capable of understanding the intention of the instructions and generates better summaries in the second and third trials. We note that the controllable generation technique can be further investigated in various evaluation settings to enhance performance.

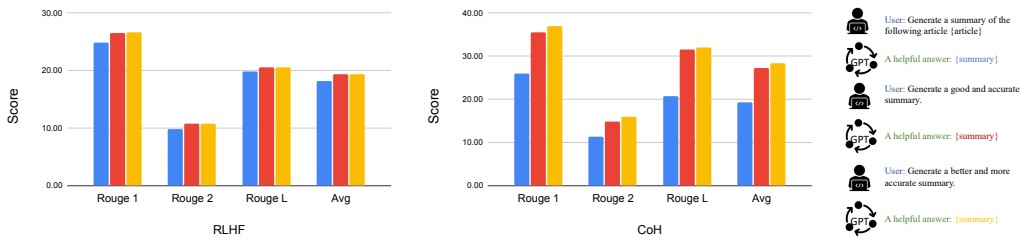

Figure 8: **Controllable generation**. **(left)**: RLHF cannot follow instructions to generate improved summary. **(middle)**: After finetuning on CoH, the model follows instructions to achieve controllable generations. **(right)**: First instruction is standard, while second and third instructions ask for improved summaries.

## E.2    ALIGNMENT TAX

We conducted an evaluation on a diverse set of few-shot tasks that are commonly used in previous studies (Brown et al., 2020; Wang and Komatsuzaki, 2021) to assess the effectiveness of aligning models with human preferences. We use Language Model Evaluation Harness[5] for evaluation. The results are reported in Table 7. Interestingly, we found that the average performance of models that were finetuned using SFT decreased after alignment. This decrease could be attributed to the issue known as *alignment tax* in language models (Ouyang et al., 2022), which underscores the importance of human evaluation (Lee et al., 2022). On the other hand, our proposed method, CoH, showed moderate improvements over both the pretrained model and supervised fine-tuned model. This result suggests that CoH is less susceptible to the *alignment tax* issue.

## E.3    COMPARISON AGAINST CHATGPT DISTILLATION

The open-source human preference datasets utilized in this study are curated based on human preferences for model generations. Although these preferences offer valuable learning signals as we have demonstrated in the experiments, the models responsible for these responses are notably less capable than proprietary models like ChatGPT. As a result, the data quality from these open-source datasets falls short when compared to conversations between ChatGPT and users which is shared online on ShareGPT. Given that the ShareGPT data showcases superior quality and greater diversity than the open-source datasets, we are interested in how our approach CoH performs when applied to open-source human preference datasets, in comparison to the SFT approach used on ShareGPT data. To this end, we compared with Koala (Geng et al., 2023) which involves supervised finetuning LLaMA (Touvron et al., 2023) using ShareGPT data. It's worth noting that we maintained consistency in the model

---

[5]https://github.com/EleutherAI/lm-evaluation-harness

Table 7: **Alignment Tax on Few-Shot Benchmark**: The results of our experiments on few-shot NLP benchmarks using the `Language Model Evaluation Harness` are presented in Table 7. We follow the same setup as in previous work (Brown et al., 2020; Wang and Komatsuzaki, 2021), including the splits for each task. The reported numbers for GPT-J are taken from its original paper, while the numbers for other models are reported by us. We average the results over 5 random seeds.

| | Zero-shot | | | One-shot | | | Few-shot | | |
|---|---|---|---|---|---|---|---|---|---|
| **Task** | GPT-J | SFT | CoH | GPT-J | SFT | CoH | GPT-J | SFT | CoH |
| ANLI R1 | 34.00 | 33.50 | 33.80 | 33.50 | 33.50 | 33.60 | 32.70 | 32.60 | 32.70 |
| ANLI R2 | 32.00 | 32.00 | 32.10 | 34.40 | 34.10 | 34.20 | 33.90 | 34.20 | 34.10 |
| ANLI R3 | 34.00 | 34.30 | 36.80 | 34.80 | 34.60 | 36.90 | 35.40 | 35.60 | 36.80 |
| ARC-C | 27.00 | 26.80 | 27.60 | 32.20 | 32.50 | 33.80 | 33.10 | 33.50 | 34.20 |
| ARC-E | 54.30 | 54.20 | 54.40 | 62.80 | 62.50 | 62.50 | 66.50 | 66.50 | 66.50 |
| BoolQ | 58.50 | 61.50 | 61.30 | 57.20 | 57.10 | 58.10 | 42.50 | 42.30 | 42.90 |
| CB | 41.10 | 41.00 | 40.50 | 41.10 | 41.10 | 40.50 | 42.90 | 42.10 | 42.00 |
| COPA | 71.00 | 70.50 | 69.90 | 80.00 | 80.10 | 80.50 | 82.00 | 82.20 | 81.50 |
| HeadQA | 23.50 | 23.00 | 23.80 | 24.00 | 23.80 | 24.30 | 23.90 | 22.50 | 22.80 |
| HellaSwag | 42.60 | 42.30 | 42.00 | 46.20 | 46.10 | 46.10 | 46.10 | 46.00 | 46.70 |
| MultiRC | 3.00 | 3.10 | 4.10 | 6.50 | 6.70 | 7.40 | 6.60 | 6.90 | 7.50 |
| ReCORD | 85.80 | 85.60 | 85.60 | 86.20 | 86.00 | 86.40 | 58.60 | 58.80 | 58.60 |
| RTE | 51.20 | 50.50 | 50.00 | 55.60 | 55.50 | 55.90 | 52.00 | 52.00 | 52.00 |
| WiC | 45.00 | 45.00 | 45.00 | 44.50 | 44.20 | 44.10 | 50.00 | 50.50 | 50.00 |
| WSC | 36.50 | 36.90 | 42.80 | 37.50 | 38.10 | 43.70 | 35.80 | 37.60 | 41.30 |
| LAMBADA (openai) | 5.50 | 5.70 | 5.70 | 5.30 | 5.40 | 5.40 | 2.50 | 2.70 | 3.60 |
| LAMBADA (standard) | 2.10 | 0.90 | 0.90 | 3.00 | 2.20 | 1.90 | 3.20 | 3.30 | 3.30 |
| LogiQA | 21.50 | 20.00 | 20.00 | 20.70 | 20.90 | 20.90 | 19.00 | 20.60 | 20.10 |
| WinoGrande | 49.70 | 50.40 | 51.20 | 50.70 | 51.80 | 53.50 | 50.70 | 51.10 | 52.80 |
| SciQ | 86.40 | 86.00 | 86.00 | 89.10 | 89.10 | 89.10 | 54.00 | 55.00 | 55.00 |
| OpenBookQA | 16.00 | 16.20 | 15.40 | 16.80 | 16.70 | 16.70 | 20.80 | 20.90 | 21.10 |
| PIQA | 72.40 | 72.40 | 72.00 | 73.60 | 73.70 | 73.50 | 74.20 | 74.00 | 74.00 |
| **Average** | 40.60 | 40.54 | **40.95** | 42.53 | 42.53 | **43.14** | 39.38 | 39.59 | **39.98** |

and training hyperparameters for both SFT and COH when applied to open-source datasets. Additionally, we integrated CoH with Koala by finetuning both the ShareGPT and open-source datasets; here, the open-source datasets provided both positive and negative examples, while ShareGPT contributed solely positive examples. We use the same human evaluation as Koala by hiring third-party human labelers to conduct pairwise comparisons of responses generated by various models. These evaluations were based on questions sourced from a holdout set exclusive to ShareGPT. Results presented in Figure 9 reveal that our approach CoH is on par with Koala in performance. Moreover, the combined approach of CoH +Koala show slightly better performance than Koala based on human ratings. Meanwhile, both C-SFT (conditional SFT) and SFT lag behind Koala considerably. This underscores the efficacy of CoH in leveraging human preferences for learning.

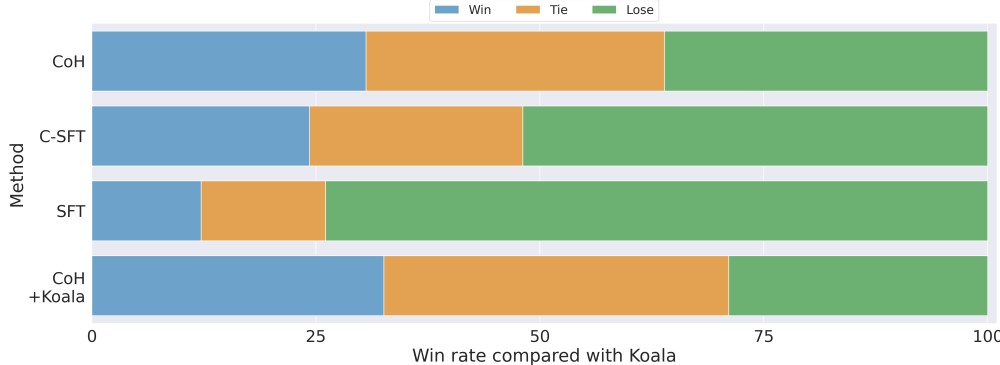

Figure 9: Evaluating various approaches with open source human preference datasets in comparison to ShareGPT's supervised finetuned Koala.

# F    Qualitative Examples

Table 8 and Table 9 show qualitative examples of summaries generated by GPT-J and CoH finetuned GPT-J. The examples are sampled from the validation split of dataset from Stiennon et al. (2020) which is based on TL;DR Reddit dataset (Völske et al., 2017).

| Example 1 |
| --- |
| Text |
| I don't know how to ask this without it sounding really silly, but this is a genuine problem. I created a throwaway because of too much personal detail in my main account. My boyfriend and I are both grad students and have been together for four months now. I absolutely love him. He's sweet, caring, tolerant, liberal, basically every single thing on my 'checklist', if I had one, is there. The problem, though, is he talks VERY slowly. I mean his speed is really slow. If I ask him anything, he takes about 3 seconds before he starts to talk, then when he starts, the words go at a very slow pace,and he even pauses for 3-5 seconds in the middle of his sentence. I get really impatient and sometimes just feel like telling him to hurry up (of course I don't, but I really feel like doing so). I asked him today, very gently, why he spoke so slow, and he said he just gets thoughts very slowly in his brain. He then went on to say he's just generally slow at thinking. I felt really sad when he said this, I don't know why. He's a Math PhD student,and we are at a pretty good school, and his research has been going slow for a while now (but this is totally normal in grad school), so I think that response was kind of a self-deprecating one since he's already low on self-esteem. Anyway, my point is, his speed of talking is just a part of him, it can't be changed. I love him so much otherwise but I just want to know how to deal with this. Conversation is so important in a relationship! How do we manage? He's an introvert and talks very little in general; I suspect this has something to do with his pace of talking? |
| GPT-J summary |
| A woman is upset because her boyfriend talks too slowly, and she doesn't know what to do. She is bothered by his slow speech, and she thinks it might be due to his low self-esteem and introverted nature. |
| CoH-tuned GPT-J |
| A woman is in love with her boyfriend who meets all her criteria except he speaks very slowly. He explains that he gets thoughts slowly and is slow at thinking. She wants to know how to manage this in their relationship as conversation is important to her. |

| Example 2 |
| --- |
| Text |
| This is my first post so please be kind :) I know that lots of people often feel confused when they come out of a long-term relationship. They think they have forgotten how to be single, or how to flirt/date. I am one of these people. The problem is, my relationship started when I had just turned 16. I have never been single - as an adult. That might sound silly. But the only time I have ever flirted or dated was as an over-confident, hormone-riddled teenager. Now I have a pretty demanding job, responsibilities blah blah... And I just don't know how to this! I'm no way in a rush to get into a new relationship, but that doesn't mean I want to be completely alone in the mean time. If anyone has experienced anything similar, or just generally has some advice, it would be greatly appreciated! |
| GPT-J summary |
| Someone is struggling after coming out of a long-term relationship that started when they were 16. |
| CoH-tuned GPT-J |
| A person is seeking advice after coming out of a long-term relationship that began when they were 16 years old. They feel confused about being single as an adult and are looking for tips on how to flirt and date again, as they don't want to be completely alone during this period. |

Table 8: Qualitative examples of GPT-J and CoH tuned GPT-J on the summarization benchmark. The input texts are sampled from the validation split of the dataset from Stiennon et al. (2020), which is based on the TL;DR Reddit dataset (Völske et al., 2017).

| Example 3 |
| --- |
| **Text** |
| Just wanted to say sorry again for wasting your time helping me out! I know I'm probably blind or something. > . < I read the FAQ, and I was able to do some of the things to get started but many of the things I am still confused on. I was able to calculate my BMR (By the way is there one calculator that is better than the others? I got a few different numbers.) However, when I went to the link on the FAQ to calculate my BF, the page was gone. Is this necessary? However, I know even without my BF that I would be classified as "skinny fat" at 5'9 and 156 pounds with a decent layer of fat. The next part I am confused on is diet. Most of the provided resources provide diets to try don't seem like they would lend themselves well to everyday/lifetime eating. I don't know how to eat healthy. The last time I tried to truly get in good shape I did it all wrong. I'm talking eating only 800-900 calories a day, counting obsessively, cutting out all my favorite foods, and interspersed with bouts of binging on aforementioned foods. Today, counting calories freaks me out because I fear that I will go back to that. I am trying to get into real fitness, like going to a gym or bodyweight fitness, but I cannot afford a gym membership and I can't seem to find something study enough to use as a pullup bar for bodyweight. What are my other options because I do want to build muscle. My final question is that I have no idea what I am truly aiming for. I want to lose my layer of fat and build muscle, but I am not sure if there's a goal weight involved or what sort of time table I am looking at or how many calories I would need to eat. This brings us full circle as I restate that I have no idea what I am doing. |
| **GPT-J summary** |
| The individual is expressing regret for taking up the reader's time in attempting to assist them. They are unable to financially support a gym membership and lack a durable apparatus for executing bodyweight exercises, such as pull-ups. |
| **CoH-tuned GPT-J** |
| The person is apologetic for taking up the reader's time and expresses confusion about some of the steps involved in getting started with a fitness journey. They were able to calculate their BMR but couldn't find a working link to calculate their BF%. They feel they are "skinny fat" and are unsure about how to eat healthily without resorting to obsessive calorie-counting. They can't afford a gym membership or a sturdy pull-up bar for bodyweight exercises. They are also uncertain about their goals for losing fat and building muscle, including their target weight, time frame, and caloric intake. |

| Example 4 |
| --- |
| **Text** |
| This happened a couple of hours ago, and i'm still a bit nervous about all this. I'm what most would consider a hardcore gamer, and as most people into this, they usually cherish and collect their favorite games and such. Today I skipped work because i woke up feeling like something had banged on my left side, hurting all over (ear infection, it's a reoccurring thing, I need to get this checked more often than not.) and decided to stay home and rest a bit more. I stayed home all day and at about 6 my dad rolls around to pick me up to go to the ER to see what's going on with my ear. During the following hour and a half I spent on the ER to get this checked and pick up the prescription the doctor gave me, I get home and find some clothing on my porch, lying on the path to my entrance. My dog was feasting on some plastic bag and at first I thought it was just some kids who'd have thrown a bag of old clothing onto my yard. As soon as I get up to my room though, things took a dire turn for me. I froze, looking baffled at what was right in front of me. My monitor was all crooked and the cables were off, and I look to the left and see this gaping hole where all my consoles were. Then it finally sunk into me: Someone robbed my house. **[Some words have been omitted because they may appear inappropriate.]** |
| **GPT-J summary** |
| A gamer skipped work due to an ear infection and got robbed. The shock and disbelief of the situation sink in, leaving the person feeling violated and anxious about the loss of their cherished belongings. The incident serves as a reminder of the vulnerability and unpredictability of life, leaving them with a sense of unease and distrust towards others. |
| **CoH-tuned GPT-J** |
| A gamer stayed home from work due to an ear infection and later went to the ER. Upon returning home, they discovered their house had been robbed and their game consoles were missing. |

Table 9: (Cont. from previous page) Qualitative examples of GPT-J and CoH tuned GPT-J on the summarization benchmark. The input texts are sampled from the validation split of the dataset from Stiennon et al. (2020), which is based on the TL;DR Reddit dataset (Völske et al., 2017).

