# OpenReview forum: "Chain of Hindsight aligns Language Models with Feedback"
_ICLR.cc/2024/Conference — ICLR 2024 poster_

### Official Review · Reviewer_us9M · 2023-11-01

**Soundness:** 4 excellent
**Presentation:** 4 excellent
**Contribution:** 3 good
**Rating:** 6
**Confidence:** 4

**Summary:**

This work proposes to learn the human preference for a LLM by training the LLM on sequences that consist of both positive and negative responses with human feedback. The results show that this method can even outperform the PPO method.

**Strengths:**

1. This paper is well-written and easy to read.
2. The proposed method is important, which can help align the fine-tuned LLM better with human preference. The presented experimental results on both summarization and dialogue tasks can validate the effectiveness of the proposed method.
3. The presented method is novel.

**Weaknesses:**

1. The largest model tested is GPT-J, which is a 6B model. It would be great that larger and more latest LLMs can be also tried, such as LLAMA-2.
2. In page 4 for the "Training" section, it is mentioned that "the model can simply ‘copy’ the example without learning to understand the underlying task". To circumvent this, this work proposes to "randomly mask between 0% and 5% of past tokens during training". What do the "past tokens" refer to? How is this masking ratio determined? Have you done some tuning on this hyper-parameter?
3. For the above point 2, this work proposes to "added a regularization term which maximize the log likelihood of the pretraining dataset". Could you specify how much pretraining data you need to use for such regularization? What is the added overhead to the training process?
4. For the dialogue task, it would be great that besides the human evaluation, another automatic evaluation based on GPT-4 can be also provided. This work was first out in this Feb or Mar., at that time, we have tried replicating this method using GPT-J and Anthropic-HHH dataset and this provided code base by the authors. We have used the same setting except that we were not using the regularization term, but we have found that the resulted model tends to generate long responses with weird grammar and repetitions. That is, the generation quality is pretty poor.
5. During inference, could you control the model to just generate the good response or the model would generate both good and bad responses and extra post-processing is needed? I doubt the case is the latter one based on my intuition and our replication experiments, which would make the inference latency double compared with other RLHF methods.

**Questions:**

1. Could you provide some qualitative examples for the model inference outputs?
2. Could you help provide the review feedback from ICML 2023 and your revision based on the reviews?

---

> ### Author Response · Authors · 2023-11-20
> **Response to reviewer us9M**
>
> Dear Reviewer us9M,
>
> Thank you so much for your positive review and insightful comments. We appreciate your feedback and have incorporated your suggestions into the revised version to improve it.
>
> **#1: Regarding your suggestion on larger and more recent LLMs**
>
> We have conducted experiments using the latest 7B LLaMA model. In our study, we compared SFT, RLHF, and CoH against Koala, a state-of-the-art LLaMA-based chatbot, serving as a robust calibration baseline. Our comparison (see Figure 9) of these methods applied to LLaMA on HH and summarization datasets with Koala used pairwise human evaluation. Notably, Koala is developed from the ShareGPT dataset, known for its superior quality and greater diversity compared to open-source datasets like HH and summarization. The results indicate that our CoH approach performs on par with Koala, and when combined (CoH + Koala), it slightly outperforms Koala based on human ratings. However, both C-SFT (conditional SFT) and SFT significantly lag behind Koala, highlighting the effectiveness of CoH in leveraging human preferences for learning.
>
>
> **#2: Clarification on "past tokens" and masking ratio**
>
> By “past tokens,” we refer to the causal tokens visible to a given token. Our approach randomly masks past tokens to prevent overfitting during training. The masking ratio was determined through preliminary experiments; we found that a larger ratio hindered training efficiency, while a smaller one made negligible difference.
>
> **#3: On regularization and pretraining data**
>
> For the regularization, we used approximately the same number of tokens as the human feedback data. This approach resulted in about a 40% increase in wall clock time for training. We will clarify this further in our revision.
>
>
> **#4: Suggestion for GPT-4 based evaluation**
>
> While we primarily focused on human evaluation for its reliability, we acknowledge its limitations in terms of scalability and expense. We have conducted a GPT-4 evaluation for LLaMA-trained SFT, PPO-RLHF, and CoH, following the MT-Bench evaluation framework. Our results show that CoH achieves the highest score, significantly outperforming SFT and PPO. We will include results of GPT-4 based evaluation for more tasks in the revision.
>
> | Model | Turn     | Score |
> |-------|----------|-------|
> | SFT   | 1        | 5.34  |
> | PPO   | 1        | 5.82  |
> | CoH   | 1        | 6.03  |
> | Model | Turn     | Score |
> | SFT   | 2        | 4.23  |
> | PPO   | 2        | 4.56  |
> | CoH   | 2        | 4.87  |
> | Model | Average  | Score |
> | SFT   |          | 4.79  |
> | PPO   |          | 5.19  |
> | CoH   |          | 5.45  |
>
> We appreciate your efforts in using our open-source code. The issue you encountered with model generation quality could be due to the absence of regularization, leading to overfitting, or incorrect hyperparameter settings. We plan to enhance our codebase with more detailed instructions and example scripts to aid future research and ease of use.
>
> **#5: Does inference process need post-processing**
> No extra post-processing is required during inference. The model is designed to generate quality responses directly. However, we are exploring potential post-processing methods that might include using search algorithms to select the best outputs or combining various token outputs for exploration. These are promising directions for future research.
>
> **#6: previous submission review feedback and revisions**
> In our ICML 2023 submission, reviewers commended the novelty of our work in utilizing pretrained language models aligned with human feedback, as well as the strong empirical results in summarization and dialogue tasks. A primary concern was the absence of an RLHF baseline, which we have now addressed in this submission. Our updated comparison includes both automatic metrics and human evaluation, showing our model's substantial superiority over RLHF.
>
> **#7: Qualitative model inference examples**
> We have included examples of model inference outputs in the appendix and plan to add more in the camera-ready version.
>
>
> Once again, thank you for your valuable feedback.

---

### Official Review · Reviewer_KdNj · 2023-11-01

**Soundness:** 3 good
**Presentation:** 3 good
**Contribution:** 4 excellent
**Rating:** 6
**Confidence:** 3

**Summary:**

This paper presents Chain of Hindsight (CoH), a technique to align LMs by converting pairwise preference data into natural langauge sequences. The proposed method involves: (1) converting preference data (context, response-good, response-bad) from different sources into natural language sequences (e.g., you are a helpful assistant. good: {good response}, bad: {bad response}), (2) training an LLM via SFT with a mask on the non-response tokens, (3) prompting the model at inference time with just 'Good:'. The intuition is that expressing natural language feedback (and examples of bad/good responses), will allow the model to learn a better notion of response quality.

Training is conducted on three datasets: WebGPT, Anthropic-hh, Summarization. Automatic evaluation is conducted on both summarization (rouge on tldr dataset) and helpful dialog (by using LM as preference model on Anthropic-hh dataset). Human evaluation is conducted for both summarization (accuracy, coherence, coverage) and helpful dialog (helpful, harmless). Across all evaluations, CoH is shown to outperform several strong baselines, including: SFT, conditional SFT (grounded on 'good'/'bad'), SFT with unlikelihood on bad responses, RLHF.

The paper also presents analysis of (1) the effect of model size (0.5B - 6B), showing that CoH scales well and (2) the impact of natural language templates (vs just having good/bad responses) demonstrating that the natural language feedback helps but that 'CoH w/o lang' strongly outperforms RLHF.

**Strengths:**

- CoH is well-motivated, interesting, novel, and performs well. The training methodology is simple, yields strong benefits and is easily extendible.
- The experiments are thorough, several datasets/settings are explored, there are strong baselines, and human evaluation is conducted.
- The analyses (e.g., w/o lang ablation, scaling) are meaningful, sound, and an impactful contribution.

**Weaknesses:**

1. More details needed about RLHF. At present, it's unclear why CoH outperforms RLHF. Is the trained RM ineffective at modeling preference (add RM performance to Fig3)? Or is the learning algorithm unable to leverage the RM effectively (could it be the choice of prompts used for RLHF?), in which case an alternate RM-based baseline could be considered (e.g. rejection sampling, reinforced self-training)?

2. Why do you only prompt with 'Good:' at inference time? It seems that an advantage of the different natural language feedback templates is that the CoH-trained LM will (hopefully) learn to *consider an initially generated bad response, and leverage it to produce a good response*. Perhaps leveraging this capability, and giving the LM an attempt to produce a better response, may yield further benefits. Some discussion or analysis about the potential of different natural language feedback templates at inference time would be useful.

3. More discussion is warranted about the performance relative to C-SFT. If my understanding is correct that C-SFT is essentially training with ("good: {good response}", "bad: {bad response}"), then I wonder if CoH with only ("good: {good response} bad: {bad response}", "bad: {bad response} good: {good response}" is expected to perform better (under the current inference setup)? If yes, why is this the case? If not, where is the advantage of CoH relative to C-SFT coming from?

**Questions:**

1. I'm not certain I understood this properly: Does 'CoH w/o lang' mean that the template will always be something like "Good: {good} Bad {bad}"? The text mentions "fine-grained language feedback" and it's not whether this is referring to the feedback templates or something else.

2. Do you have insights on how this method would scale from pairwise comparisons to N-way rankings? I thought that WebGPT had >2 responses per prompt (but might be mistaken) -- if so, it would be great to understand how this was handled.

3. Is the loss masking during CoH training really important? I would not have expected it to make a substantial difference.

*See additional questions in weakness section*

---

> ### Author Response · Authors · 2023-11-20
> **Response to reviewer KdNj**
>
> Dear reviewer KdNj,
>
>
> Thank you so much for your positive review and insightful comments. We appreciate your feedback and suggestions, which we have incorporated into the revised version to improve it.
>
> **#1: More discussion on RLHF**
>
> PPO-based RLHF suffers from reward over-optimization, as indicated in [1]. This paper also suggests training a very large model to address this issue. Despite experimenting with large models (up to 13B parameters due to compute constraints) for our PPO RLHF, we observed no significant improvements. Our hypothesis is that an even larger-scale reward model might be necessary. Additionally, we have tuned the PPO pipeline using the HuggingFace TRL library and experimented with varying the pretraining data regularization term for PPO RLHF training. Due to the complexity of reward learning and PPO RL training with multiple models, achieving optimal performance is challenging. CoH's advantage lies in its likelihood-based training, which aligns with the pretraining objective, thereby facilitating tuning and ensuring effective performance. We anticipate that future research will further explore the strengths and weaknesses of PPO RLHF and CoH, potentially leading to improved training algorithms.
>
> [1] Scaling Laws for Reward Model Overoptimization by Leo Gao, John Schulman, Jacob Hilton
>
>
> **#2: On using 'Good:' prompts at inference time**
>
> We agree that prompting CoH-trained LM with other instructions at inference time could improve performance. To this end, we have conducted experiments to compare prompting model with `Good:`, and prompting model with `Good:` followed by another round of `Better:`. The results shown in Figure 8 shows that CoH-trained LM can learn to consider the initially generated response and improve upon it.
>
>
> **#3: More discussions about the performance relative to C-SFT.**
>
> CoH differs from C-SFT in that it allows the model to reference a bad response when predicting a good one, and vice versa. This approach is crucial as it enables the model to learn directly from feedback and examples. As shown in experiments, CoH outperforms C-SFT substantially.
>
>
> **#4: Clarification on 'CoH w/o lang'**
>
> By this, we mean consistently training with the "Good: {good} Bad: {bad}" template. The term “fine-grained language feedback” refers to the use of a variety of task-dependent language templates, such as "A good summary: {good}, a worse summary: {bad}" and "You are a helpful assistant: {good}, you are an unhelpful assistant: {bad}". These templates are detailed in the box at the bottom of page 3 and in Table 6. We will clarify this in the revised manuscript.
>
>
> **#5: On scaling to N-way rankings**
>
> CoH can be readily adapted for N-way rankings by structuring N-way ranked outputs into a sequence. However, a potential challenge arises if N is significantly large, as this would proportionally increase the context size. Regarding the WebGPT dataset, each prompt includes two answers, which we rank according to the accompanying scores. We will provide more details about this in our revision.
>
>
> **#6: Importance of loss masking during CoH training**
>
> Loss masking is beneficial for stabilizing training, as predicting control tokens can introduce noise into the prediction of desired outputs. It also aids in regularizing the model and preventing overfitting to specific training examples. While CoH is effective without loss masking, we included it as it enhances performance without additional computational costs.
>
> Once again, thank you for your valuable feedback.

---

### Official Review · Reviewer_WQPV · 2023-11-01

**Soundness:** 4 excellent
**Presentation:** 4 excellent
**Contribution:** 4 excellent
**Rating:** 8
**Confidence:** 3

**Summary:**

The paper introduces "Chain of Hindsight", a novel technique for fine-tuning language models using any form of feedback, regardless of its polarity. This method is designed to overcome the inefficiencies and challenges of previous approaches that either rely on handpicked, positively-rated model generations or reinforcement learning (RL) with its imperfect reward functions and optimization challenges. CoH is inspired by how humans learn from extensive feedback in languages and aims to align language models more closely with human preferences and values. CoH converts all types of feedback into sequences of sentences, which are then used to fine-tune the model. This approach leverages the language comprehension capabilities of language models, training them to generate outputs based on feedback while learning to identify and correct negative attributes or errors. The method involves conditioning the model on a sequence of model generations paired with feedback, thus enabling learning from both positive and negative feedback.

**Strengths:**

1. The Chain of Hindsight (CoH) method is a novel approach, addressing the limitations of previous methods like supervised fine-tuning  and Reinforcement Learning with Human Feedback. It's innovative in using both positive and negative feedback for model training.
2. Simplicity and Scalability: The CoH method maintains the same training objective as pretraining, which simplifies the training process and enhances scalability. This is a significant advantage over more complex systems like RLHF. The paper reports significant improvements in tasks like summarization and dialogue generation, indicating that CoH effectively enhances model performance and alignment with human preferences.

**Weaknesses:**

1. Limited Scope of Testing: The paper only considers two evaluation benchmarks such as dialogue and summarization benchmarks. It is not clear how the model performs on the standard academic benchmark. Broader testing across diverse datasets and real-world scenarios would be necessary to fully validate the approach.

**Questions:**

1. How is the method compared with the other baseline such as rejection sampling?

---

> ### Author Response · Authors · 2023-11-20
> **Response to reviewer WQPV**
>
> Dear reviewer WQPV,
>
> Thank you so much for your positive review and insightful comments. We appreciate your feedback and suggestions, which we have incorporated into the revised version to improve it.
>
>
> **#1: Relation to rejection sampling**
>
> We appreciate the question, CoH leverages in-context power of large language models to learn from a sequence of examples and feedback, while rejection sampling is an orthogonal direction that uses a reward model to select top N model outputs for finetunning. CoH and rejection sampling can be directly combined to potentially further improve the performance of CoH, which is a promising future direction.
>
>
> **#2: Broader scope of testing**
>
> We acknowledge that broader scope of testing as an area for future improvement. Our initial focus was on demonstrating the effectiveness of CoH in tasks where feedback dynamics are crucial, such as dialogue and summarization, and evaluated on the primary dataset in each category. However, we agree that broader testing across diverse datasets is essential for validating the method's applicability in a wider context. In future work, we aim to include a more varied set of benchmarks, including code generation from unit tests feedback, to better evaluate the method's generalizability and performance in different scenarios.
>
> Once again, thank you for your valuable feedback.

---

### Official Review · Reviewer_aon5 · 2023-11-01

**Soundness:** 3 good
**Presentation:** 4 excellent
**Contribution:** 4 excellent
**Rating:** 8
**Confidence:** 4

**Summary:**

This paper proposes a novel technique called Chain of Hindsight (CoH) that aims to align language models with human preferences and values by leveraging human feedback. The authors convert all types of feedback into sequences of sentences, which are then used to fine-tune the language model. By conditioning the model on a sequence of model generations paired with feedback, the model is trained to generate outputs based on feedback while learning to identify and correct negative attributes or errors. The authors evaluate the proposed approach on summarization and dialogue tasks and report significant improvements over existing baselines in both automatic and human evaluation.

**Strengths:**

(1) The paper presents a novel technique, Chain of Hindsight (CoH), which addresses the challenge of aligning language models with human preferences and values by leveraging human feedback.

(2) The approach is easy to optimize and can learn from any form of feedback, regardless of its polarity.

(3) The paper provides a well-structured review of relevant literature, including prior works on learning from human feedback and language modeling.

**Weaknesses:**

This is a good paper. I see no reasons to reject it. Only a few comments:
1) I am confused by the illustrated examples. In the Figure 1, the prompt template uses 'a helpful answer' / 'an unhelpful answer' while in the Section 1, they are using 'Good' / 'Bad'. It would be better to be consistent.

2) Some important studies [1,2] are missing. It would be better to include them and  have a discussion.

[1] RAFT: Reward rAnked FineTuning for Generative Foundation Model Alignment

[2] Direct Preference Optimization: Your Language Model is Secretly a Reward Model

**Questions:**

See Weaknesses.

---

> ### Author Response · Authors · 2023-11-20
> **Response to reviewer aon5**
>
> Dear reviewer aon5,
>
> Thank you so much for your positive review and insightful comments. We appreciate your feedback and suggestions, which we will incorporate into the revised version to improve it.
>
>
> **#1: Consistency in terminology**
>
> We acknowledge the inconsistency you pointed out in our use of terms 'helpful/unhelpful' and 'good/bad' across different sections. We agree that maintaining a consistent terminology is crucial for clarity, and we will revise our manuscript to ensure uniformity in the terms used throughout the paper. However, note that we train with a variety of natural language feedback (see Table 6 in the paper).
>
>
> **#2: Include a discussion on related studies**
>
> We appreciate your suggestion to include the studies “RAFT” and “Direct Preference Optimization”. These are indeed significant contributions to the field. We will incorporate these references into our paper and discuss how our work aligns with, and differs from, these studies.
>
> In addition to these specific changes, we will thoroughly review our manuscript once more to ensure clarity and coherence in our presentation. We believe that addressing these points will strengthen our paper and better communicate our contributions to the field.
>
>
> Once again, thank you for your valuable feedback.

---

### Meta-Review · Area_Chair_Y5Tj · 2023-12-11

**Metareview:**

The submission proposes Chain of Hindsight, in which feedback (either positive or negative) is converted to natural language sentences and used to better train models. Reviewers are overall positive on the paper, and found it novel and interesting, and that the method performs well empirically. There are no major concerns raised in the reviews, but the paper could be further strengthened by including mode tasks and larger models. Overall, this is solid work.

**Justification For Why Not Higher Score:**

Probably needs more thorough evaluation to be a spotlight.

**Justification For Why Not Lower Score:**

No major weaknesses, seems solid and interesting.

---

### Decision · Program_Chairs · 2024-01-16

Accept (poster)